# Method for Predicting RUL of Rolling Bearings under Different Operating Conditions Based on Transfer Learning and Few Labeled Data

**DOI:** 10.3390/s23010227

**Published:** 2022-12-26

**Authors:** Wei Sun, Haowen Wang, Zicheng Liu, Ronghai Qu

**Affiliations:** School of Electrical and Electronic Engineering, Huazhong University of Science and Technology, Wuhan 430074, China

**Keywords:** rolling bearing, RUL, DAAN, MMD

## Abstract

As industrial development increases, electric machine systems are more widely used in industrial production. Rolling bearings play a key role in machine systems and so the prevention of faults in rolling bearings is more important than ever before. Recently, with the development of artificial intelligence, neural networks have been used to monitor the remaining useful life of rolling bearings. However, there are two problems with this technique. First, a network trained by data for a single operating condition (source domain) cannot predict the remaining useful life of bearings under a different operating condition (target domain), such as a different load or speed. Second, a large number of labeled data are needed for network training, but the acquisition of labeled data for different operating conditions is a challenging task. To address these problems, this paper proposes a domain-adaptive adversarial network, in which a transfer learning strategy and maximum mean discrepancy algorithm are used for network optimization, so that remaining useful life can be predicted without labeled data in target domain training. Our results confirm that a model trained by source domain data alone cannot predict the remaining useful life of bearings under different conditions, but the domain-adaptive adversarial network can accurately predict remaining useful life for varying operating conditions. The method proposed also exhibits good performance even if there are noises in the signals.

## 1. Introduction

As industrial development has increased, EMSs have become more widely used in industrial applications [1,2,3]. However, if EMSs are faulty or broken, they present a threat to application processes and to the physical safety of people. Assessment of the health condition of EMSs is therefore an important matter, and this includes assessment of likely future performance.

In recent years, as artificial intelligence technology has developed, neural networks have become widely used in many fields. Reports in the research literature confirm this. In [4], a backpropagation artificial neural network was employed for compressive strength prediction of alkaline-activated slag concrete. In [5], an artificial neural network was used to identify areas flooded by a cyclonic storm. Published papers also describe various deep learning models proposed and used for EMS fault diagnosis. In [6], a network constructed by sparse filtering and a softmax classifier was used to extract features from bearing vibration signals and to identify fault types of a motor bearing at different loads by adaptive means. In trials, this method obtained a good degree of accuracy. In [7], an improved machine-learning-based fault diagnosis method with adaptive secondary sampling filtering was proposed for multiphase drive systems. By this method, open-circuit faults in power switches were effectively diagnosed. In [8], a Bi-LSTM network was used to construct a fault diagnosis model of early gear pitting. This network was well designed to extract pitting features from the raw vibration signals of gears and achieved good diagnosis accuracy. In [9], monitoring data were acquired by several vibration sensors, and these signals were used as inputs to a representation learning subnetwork, and to an RUL estimation subnetwork, to predict the RUL of bearings.

The methods just described exhibit two main limitations. First, they focus on sudden faults rather than damage caused by age. Second, because of the identical distribution assumption of deep learning, they only work when the predicting samples and training samples are from the same distribution. However, machines are routinely used in different operating conditions, meaning that a model trained by data of one condition cannot be applied directly to other conditions. In addition, obtaining extensive amounts of labeled data of different operating conditions is expensive. Training a model for different conditions is, therefore, a challenging task, and the generalization of the methods just described is weak. However, and as described in [10], most EMS accidents involve rolling bearings, and this may represent a fruitful area of future research. In this study, considering the findings reported above, we sought to make RUL predictions for rolling bearings under different operating conditions with few labeled data. Our methods and results are set out in this paper.

Transfer learning, as a deep learning algorithm, provides a solution for different distribution prediction, and has been widely used in various fields [11,12,13]. In [11], transfer learning was used in image generation. A generative adversarial network was pretrained by image datasets; then, the network was transferred to generate images in different domains. In [12], transfer learning was used for food material recognition. Recognition knowledge learned from publicly available datasets was transferred to the real-world restaurant domain, and the classifier was used to recognize food material. In [13], transfer learning was used for the short-term prediction of extreme weather events in different regions. The base CNN model was trained using radar data from near Beijing, China; the transferred models were then successfully used for prediction near Guangzhou, China. In [14], Pan and Yang reviewed developments in transfer learning. In [15], Selver et al. provided an evaluation strategy for transfer learning. In transfer learning, there are two main parts: first, the source domain $D_{s}=X_{s}$, belonging to feature space $S_{s}$ and task $T_{s}$; second, the target domain $D_{t}=X_{t}$, belonging to feature space $S_{t}$ and task $T_{t}$. Generally, the source domain samples are labeled, but there are either few labeled samples in the target domain or none at all. Transfer learning aims to improve the performance of task $T_{t}$ in the target domain, with the help of information from the source domain.

When predicting the health of bearings, two different operating conditions can be regarded as a source domain and target domain, respectively, so that the feature spaces and tasks are the same for both source and target domains, i.e., $S_{s}=S_{t}$ and $T_{s}=T_{t}$. However, because of the different operating conditions, the marginal distributions of the two domains are different, i.e., $P\left(X_{s}\right)\neq P\left(X_{t}\right)$. Some published works on transfer learning are especially relevant in this regard. In [16], a subdomain adaptation transfer learning network was established to predict different bearing faults. By this method, class and domain misalignment issues in the fault diagnosis of bearings were overcome, and good performance was obtained. In [17], a stack auto-encoder transfer learning algorithm based on the class SAE-CSDF was proposed to address the “few data” problem in diagnosing bearing faults; experimental results demonstrated that this method improved the accuracy of diagnosis.

In the above-described transfer learning methods used for bearing fault diagnosis, the data of different fault types are distributed differently, while data of similar fault types are distributed similarly. Accuracy can therefore be improved just by reducing the differences between the fault types. However, when predicting the RUL of bearings, the label is a real number instead of a class, so it is hard to apply these methods directly to the task of RUL prediction. In order to predict the RUL of bearings under different operating conditions, an RUL prediction method based on transfer learning is proposed in this paper. The originality of the method can be summarized as follows:(1)A DAAN is used to address the problem that a network trained by data under one condition cannot then be used for different conditions. Transfer learning and MMD are employed to reduce the differences in vibration signals under different operating conditions, so the network can be used under different conditions.(2)To train the network, labeled data are needed for only one operating condition. For other conditions, unlabeled data are used. This eliminates the difficulty of obtaining large quantities of labeled data of different operating conditions.(3)Compared with the results of similar previous works, the method proposed here results in more accurate performance, with fewer data to train.

The details of the proposed method and the experimental results are described in the remainder of this paper.

## 2. Proposed Method

Figure 1 shows the structure of the proposed method. A DAAN is used for predicting the RUL of bearings under different operating conditions. In the DAAN, the CNN [18] is used to extract features from the samples, and a fully connected network is used to predict RUL. An adversarial transfer learning strategy and MMD are used to train the network and to determine the RUL prediction under different operating conditions. Details of each stage in the process are as follows:

### 2.1. Signal Preprocessing

For the network, the original data cannot be used directly, and preprocessing is necessary. To fit CNN, the signals need to be processed and constructed as two-dimensional samples.

WPT is a method based on multiresolution theory and orthogonal wavelet algorithms. In WPT, the signals are decomposed to different frequency bands by a series of high- and low-pass filter matrices. After the wavelet function is determined, the coefficients of these filter matrices are obtained. In practical engineering, WPT is usually used to process nonstationary signals, such as vibration signals [19,20]. WPT is described in detail in [20]. In this paper, WPT is used to transform the vibration signals to a time–frequency domain, which is a two-dimensional space, so that samples can be used as input to the network.

A whole-life dataset $V={\left\{v^{i}\right\}}_{i=1}^{m}$ contains *m* vibration signals. A larger superscript indicates a lower corresponding RUL. For a vibration signal $v^{i}\in R^{N\times 1}$, this can be regarded as a wavelet coefficient $a_{1}^{0}$ at level zero; wavelet coefficients of each successive level can then be obtained as in (1), following [19]:(1)$$\{\begin{array}{l} a_{2j-1}^{n}=G^{n}\cdot a_{j}^{n-1}\\ a_{2j}^{n}=H^{n}\cdot a_{j}^{n-1} \end{array}\left(j=1,2,\ldots ,2^{n-1}\right)$$
where $a_{j}^{n}$ is the $j^{th}$ wavelet packet coefficient of the $n^{th}$ level, $H^{n}$ and $G^{n}$ are the high- and low-pass filter matrices of the $n^{th}$ level, which are $N/2^{n}\times N/2^{\left(n-1\right)}$ dimensional. The feature vector $f^{i}$ of the vibration signal is made up by summing and L2-normalizing wavelet coefficients for each level, as in (2):(2)$$f^{i}=\frac{\left[\sum a_{1}^{n}\;\sum a_{2}^{n}\;\ldots \;\sum a_{2^{n}}^{n}\right]}{\sqrt{{\left(\sum a_{1}^{n}\right)}^{2}+{\left(\sum a_{2}^{n}\right)}^{2}+\ldots +{\left(\sum a_{2^{n}}^{n}\right)}^{2}}}$$

To construct the two-dimensional samples for CNN, and ensure the samples are square matrices, a total of $2^{n}$ feature vectors with similar RUL values are used to construct the sample $s^{i}$ as in (3):(3)$$s^{i}=\left[\begin{array}{c} f^{i+1-2^{n}}\\ \;\vdots \\ \;\;f^{i-1}\\ \;\;\;f^{i} \end{array}\right]$$

Finally, the samples are used as input to the network.

### 2.2. Network Training

In this paper, bearings under one operating condition are regarded as source domain samples $X_{1}={\left\{x_{1}^{i}\right\}}_{i=1}^{m_{1}}$. The RULs of the source domain samples construct $Y_{1}$. Bearings under different operating conditions are regarded as target domain samples $X_{2}={\left\{x_{2}^{i}\right\}}_{i=1}^{m_{2}}$. In addition, *L* represents domain labels (source: 0, target: 1) and the labels represent whichever domain the samples belong to. The diagram of network training is shown in Figure 2. The specific training steps are as follows (it should be pointed out that the weight-shared networks update at the same time in the training).

Step A

In this step, Net1 and Net2 are fixed, and the loss function $Loss_{d}$ is minimized by updating Net3. $X_{1}$ and $X_{2}$ are used as input to Net1 and Net3 to predict condition labels. The classification loss ($Loss_{d}$) can be then obtained by MSE as in (4):(4)$$Loss_{d}=\frac{1}{m}\sum_{i=1}^{m}\left(l^{i}-{\hat{l}}^{i}\right)$$
where *m* is the number of samples, and $l^{i}$ and ${\hat{l}}^{i}$ are the real domain label and the predicted domain label, respectively.

Step B

In this step, Net3 is fixed, and the loss function $Loss_{g}$ is minimized by updating Net1 and Net2. $Loss_{g}$ includes three parts: regression loss, classification loss and label MMD loss, as follows:

Regression loss ($Loss_{g1}$): $X_{1}$ is used as input to predict RUL labels; these predicted labels are then combined with $Y_{1}$ to calculate $Loss_{g1}$ by MSE as in (5):(5)$$Loss_{g1}=\frac{1}{m}\sum_{i=1}^{m}\left(y^{i}-{\hat{y}}^{i}\right)$$
where *m* is the number of samples, and $y^{i}$ and ${\hat{y}}^{i}$ are corresponding real RUL labels and predicted RUL labels, respectively.

Classification loss ($Loss_{g2}$): Similarly to the generative adversarial network described in [21], $Loss_{g2}$ is set to negative $Loss_{d}$ to achieve adversarial training of Net1 and Net3. The invariant features in the domains can then be extracted by the trained feature extractor, following [22].

Label MMD loss ($Loss_{g3}$): Due to the lack of target domain labels, the unlabeled samples in the target domain cannot be used to train the parameters in Net2 [23]. However, because the samples from the source domain and the target domain are both collected from the whole life of the bearings, their RUL labels should obey the same distribution. MMD can then quantify the difference of the distributions, by calculation, as in (6) [24]:(6)$$MMD\left(X_{1},X_{2}\right)=\frac{1}{m_{1}^{2}}\sum_{i=1}^{m_{1}}\sum_{j=1}^{m_{1}}k\left(x_{1}^{i},x_{1}^{j}\right)-\frac{2}{m_{1}m_{2}}\sum_{i=1}^{m_{1}}\sum_{j=1}^{m_{2}}k\left(x_{1}^{i},x_{2}^{j}\right)+\frac{1}{m_{2}^{2}}\sum_{i=1}^{m_{2}}\sum_{j=1}^{m_{2}}k\left(x_{2}^{i},x_{2}^{j}\right)$$
where *k*(·,·) is the kernel function.

We now recall that a radial basis function can map the distribution to infinite-dimensional space, where the two distributions can be well measured. For this reason, we select a radial basis function as the kernel function as in (7):(7)$$k\left(x,x^{\prime }\right)=e^{-\frac{{∥x-x\prime ∥}^{2}}{2\sigma^{2}}}$$
where *σ* is kernel width, which represents the influence range of the radial basis function. $Loss_{g3}$ is the MMD of the predicted RUL of source domain and target domain samples; reducing this loss can update the parameters in Net3, using target domain unlabeled samples.

Finally, $Loss_{g}$ is the weighted sum of the above three parts, as in (8):(8)$$Loss_{g}=Loss_{g1}+Loss_{g2}+Loss_{g3}$$

A single repetition of Step A and Step B is regarded as one iteration. The network is trained until the threshold of iteration is attained, and the well-trained network can then be used for RUL prediction purposes.

### 2.3. RUL Prediction

A well-trained network is obtained after network training. The RUL of bearings under different operating conditions can be used by Net1 and Net2. The whole flowchart of the proposed method for RUL prediction is shown in Figure 3.

## 3. Experiment

### 3.1. Data Description and Parameters Configuration

The datasets used in the experiments can be found in [25]. The datasets include five different load conditions and seventeen whole bearing remaining life vibration signals in all. Signals are collected from vertical and horizontal directions at a sampling frequency of 25.6 kHz. Typical vibration signals are shown in Figure 4. Signals are sampled every 0.1 s for each 10 s, so each data segment has 2560 points.

The horizontal-direction vibration signals used in the experiments are listed in Table 1. Each data segment is regarded as a vibration signal $v^{i}$, and the level of wavelet packet decomposition *n* is set to five, which means each segment contains 32 wavelet packet coefficients. To construct one square matrix sample, 32 data segments with similar RUL values are used. Considering the whole-life time is different for each bearing, the RUL of each bearing is here represented as a percentage for convenience of comparison and evaluation.

For the network, the CNN is constructed by two convolutional layers and two max pooling layers. Flattened features extracted by CNN are used as inputs to the fully connected network. The outputs of Net2 and Net3 are RUL and condition labels, so the output layer size of both Net2 and Net3 is one. The kernel width σ is also set to one. The parameter set of the whole network is shown in Table 2.

### 3.2. Validity and Generalization of the Proposed Method

Generally, a trained network becomes invalid when the operating condition changes. To experimentally assess the validation and generalization of the proposed method under different operating conditions, two cases are used for testing purposes. In Case I, bearing2_1 is regarded as the source domain, bearing3_1 is regarded as the target domain, and bearing3_3 is used to test. In Case II, bearing3_1 is regarded as the source domain, bearing2_1 is regarded as the target domain, and bearing2_2 is used to test.

The results are shown in Figure 5 and Figure 6. For the method without transfer learning, only labeled samples from the source domain are used to train Net1 and Net2. It is obvious that the performance of the proposed method is much better than that of the non-transfer-learning method, and this verifies the validity of the former. In addition, the results show that the proposed method exhibits good performance in both of the two cases, demonstrating that the proposed method effectively transfers to different operating conditions.

To evaluate the effectiveness of the prediction results, the following metrics are used to measure the performance:(1)RMSE is calculated as in (9) and can be used for measuring the average absolute error.
(9)$$RMSE=\sqrt{\frac{1}{m}\sum_{i=1}^{m}{\left({\hat{y}}^{i}-y^{i}\right)}^{2}}$$

(2)MAPE is calculated as in (10) and can be used for measuring relative error.


(10)
$$MAPE=\frac{1}{m-1}\sum_{i=1}^{m-1}\frac{\left|{\hat{y}}^{i}-y^{i}\right|}{y^{i}}$$


(3)Precision [26] is calculated as in (11), and this metric can quantify the dispersion of the prediction error around its mean.

(11)$$\mathrm{Precision}=\sqrt{\frac{1}{m}\sum_{i=1}^{m}{\left(\varepsilon_{i}-\bar{\varepsilon }\right)}^{2}}$$
where *m* is the number of samples, and ${\hat{y}}^{i}$ and $y^{i}$ are the predicted RUL and real RUL of the sample $s^{i}$, respectively. The error $\varepsilon^{i}={\hat{y}}^{i}-y^{i}$, mean error $\overset{¯}{\varepsilon }=\frac{1}{m}\sum_{i=1}^{m}\varepsilon^{i}$, RMSE and MAPE are all used to measure the accuracy of prediction. The closer to zero these values are, the more accurate the RUL estimation of the proposed method. Precision reflects the dispersion of error, so when its value is close to zero, prediction errors are more concentrated, and the method is more stable. When the RUL is close to zero, the value of MAPE will be large, even if the absolute error is small. In addition, bearings are usually taken out of use when they are predicted to be broken by industrial application. For this reason, samples with an RUL of less than 10% are ignored when calculating MAPE. The results are shown in Table 3.

It can be seen that the proposed method achieves better performance on each metric than the method without transfer learning.

To further reveal the effectiveness of the transfer learning strategy, the features extracted by the CNN are analyzed visually. PCA is used for data dimension reduction, and the probability density distributions of the first principal component of the deep features are compared in Figure 7 and Figure 8, where the source domain and target domain are bearing2_1 and bearing3_1, respectively, in Figure 7, and bearing3_1 and bearing2_1, respectively, in Figure 8.

It can be seen that, without using transfer learning, the distributions of the two domains are very different, and this is obviously caused by the different marginal distributions of these two domains. When the transfer learning strategy and MMD loss are employed to reduce the differences in these two distributions, the probability density distribution curves become similar, and the invariant features of the domains are extracted. It is important that the regressor effectively predicts RUL under different operating conditions based on the extracted features. These results show the effectiveness of the transfer learning strategy for reducing the differences in the distributions and for predicting RUL under different operating conditions.

### 3.3. The Robustness of the Method

Because bearings are routinely used under extreme operating conditions in harsh working environments, the signals from bearings collected from real-world sites are more confused than those collected in experimental environments, and usually contain a great deal of interference and noise. For this reason, algorithm robustness is important.

To test the robustness of the proposed method, the experiments under Case I (source domain: bearing2_1, target domain: bearing3_1, test bearing: bearing3_3) were carried out with different noise level data to simulate extreme operating conditions. White Gaussian noise signals with a mean value of zero were added to the vibration signals of all the bearing vibration signals used in the experiments, and the noise level was determined by its standard deviation. The experiments were carried out several times under different noise levels, and the accuracy results obtained are shown in Table 4.

It can be seen that use of the proposed method results in acceptable performance until the standard deviation of the noise reaches 2 g. Considering that the standard deviation of the whole-life vibration signals of bearing2_1 and bearing3_1 are 10.40 g and 5.88 g, respectively, the noise with 2 g standard deviation can be regarded as a strong interference with the signals. It is therefore reasonable to conclude that the proposed method is robust and performs well even when the vibration signals contain noise.

### 3.4. Comparison with Related Methods

To further assess the proposed method, its performance is compared to that of previously described RUL prediction methods using transfer learning, as shown in Table 5. In [27], an RUL estimation method using DTMLKR was used. Bearings under condition1 or condition 2 were regarded as source domain, and target domain, respectively. Source domain bearings and two target domain bearings were used to train the model, and the trained model was used to predict RULs of the remaining target domain bearings. In [28], the whole RUL of the bearing was divided into a normal stage and a rapid degradation stage. An RUL prediction method based on SCAE and MK-MMD was used to estimate the rapid degradation stage RUL of bearings under condition2. In [29], the fault occurrence time of bearings was determined first using a hidden Markov model. Bearings under condition 1 were then regarded as source domain, and a novel TLMLP was used to predict the RUL of bearings under condition 2 and condition 3.

The results show that the method proposed in this paper results in good absolute accuracy, compared with the other methods, but its relative error performance is slightly poorer in some instances. Firstly, compared with the DTMLKR method, the proposed method is a little more accurate. Secondly, the SCAE method exhibits better performance on the metrics, but this method is only used to predict rapid degradation stage RUL, whereas the proposed method is used for the whole life of the bearing. This is a more challenging task than predicting only the rapid degradation stage, and this explains why the accuracy of the proposed method is lower than that of the SCAE method. In addition, the amount of source and target domain data used in the proposed method is less than that of the SCAE method, and this also contributes to the lower accuracy of the proposed method. However, this deficiency is small and acceptable. Finally, in comparison with TLMLP, the proposed method obtains better accuracy with fewer training data, indicating a clear superiority of the proposed method.

## 4. Conclusions

In this paper, the use of DAAN is proposed for predicting the RUL of bearings under different operating conditions. In the method described here, WPT is used for signals pre-processing and for the construction of two-dimensional data. A transfer learning strategy and MMD algorithm are then used for network optimization, which reduces the differences between the features extracted from samples obtained from different operating conditions samples, so the network can predict the RUL of bearings under different operating conditions.

The results demonstrate that, because of its use of a transfer learning strategy and MMD algorithm, the proposed method can map vibration data under different operating conditions to a similar distribution, so the network can predict the RUL of bearings under different conditions. In addition, the proposed method delivers good performance even if the vibration signals contain noise.

## Figures and Tables

**Figure 1 sensors-23-00227-f001:**
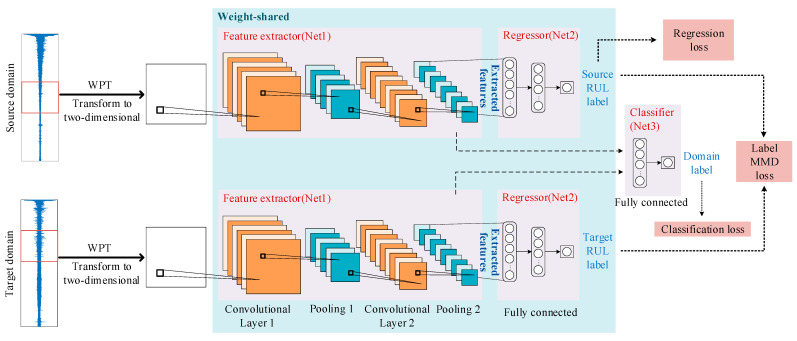
The proposed RUL prediction method.

**Figure 2 sensors-23-00227-f002:**
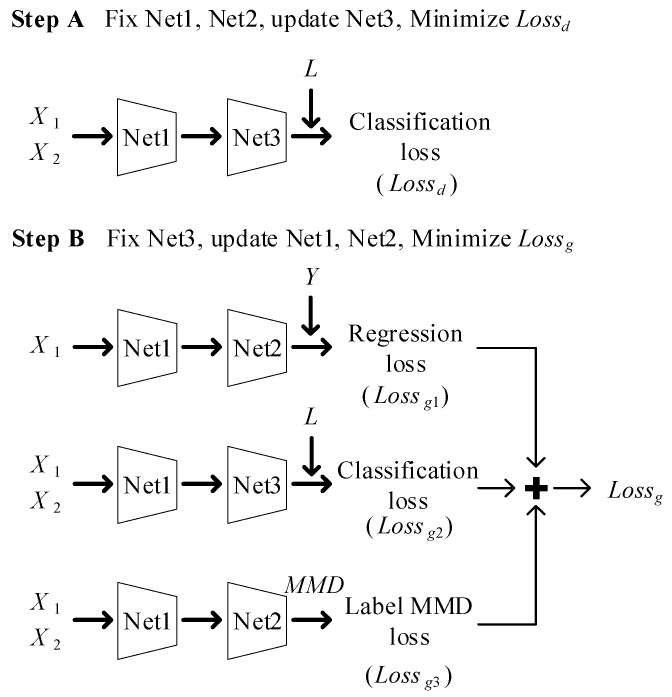
Diagram of network training steps.

**Figure 3 sensors-23-00227-f003:**
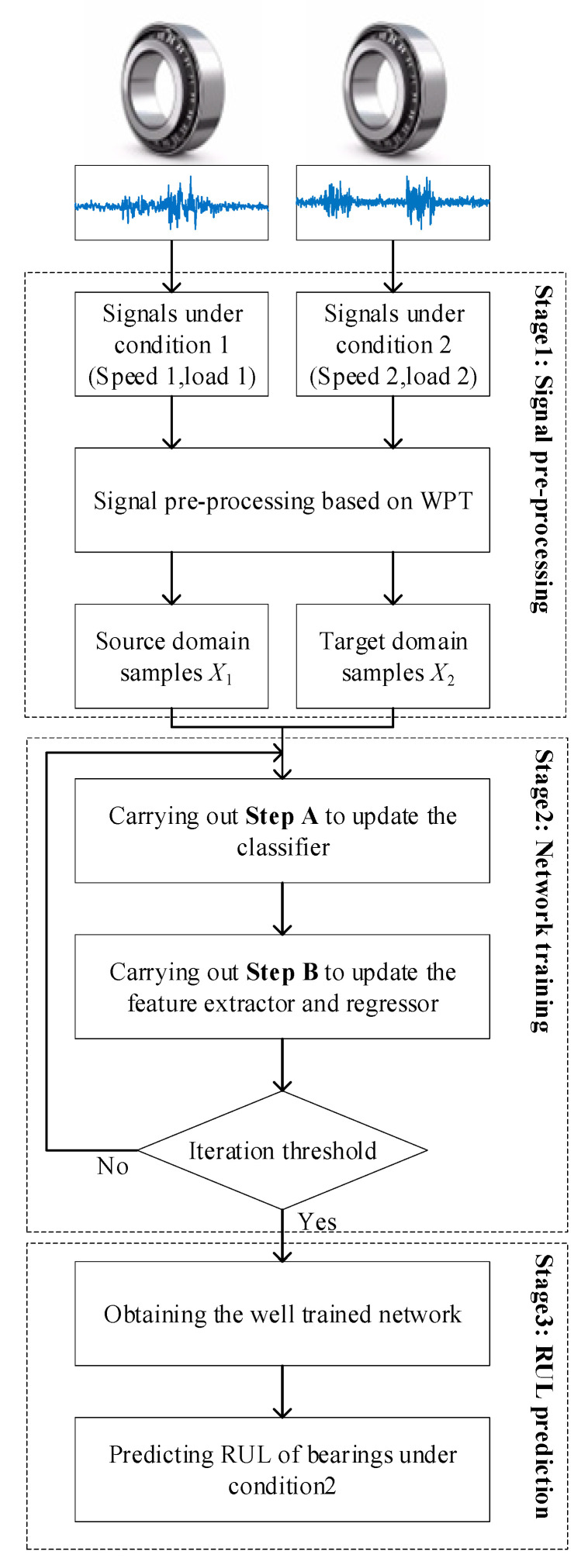
Flowchart of the proposed method for RUL prediction.

**Figure 4 sensors-23-00227-f004:**
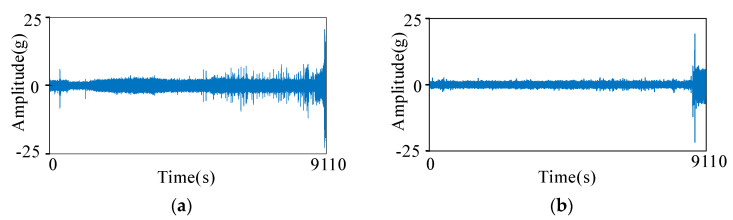
Vibration signals of bearing2-1. (**a**) Horizontal direction. (**b**) Vertical direction.

**Figure 5 sensors-23-00227-f005:**
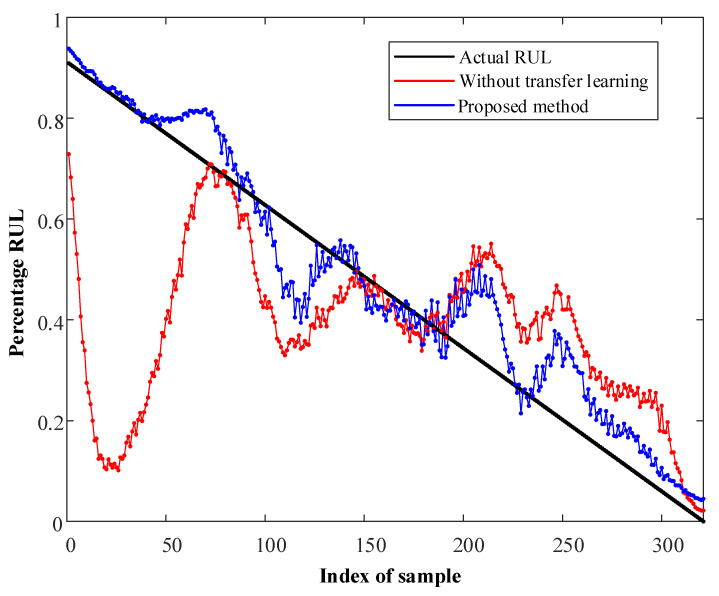
RUL prediction of bearing3_3 in Case I.

**Figure 6 sensors-23-00227-f006:**
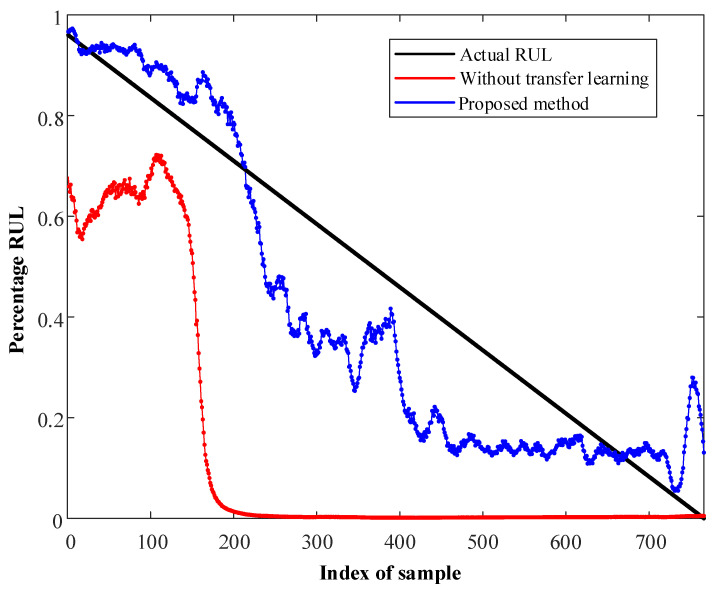
RUL prediction of bearing2_2 in Case II.

**Figure 7 sensors-23-00227-f007:**
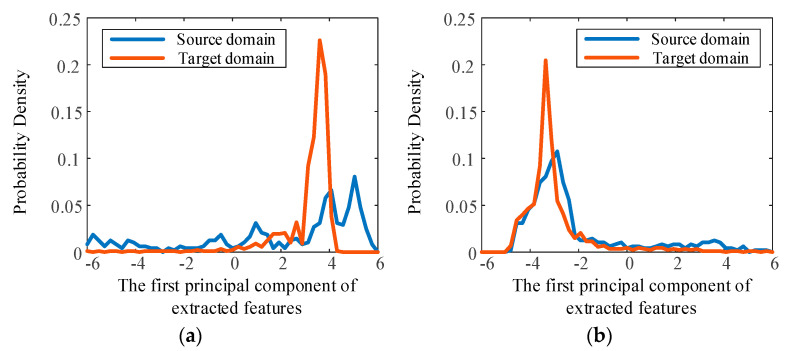
Probability density distribution of the first principal component of extracted features. Source domain is bearing2-1, target domain is bearing3-1. (**a**) Without transfer learning. (**b**) With transfer learning.

**Figure 8 sensors-23-00227-f008:**
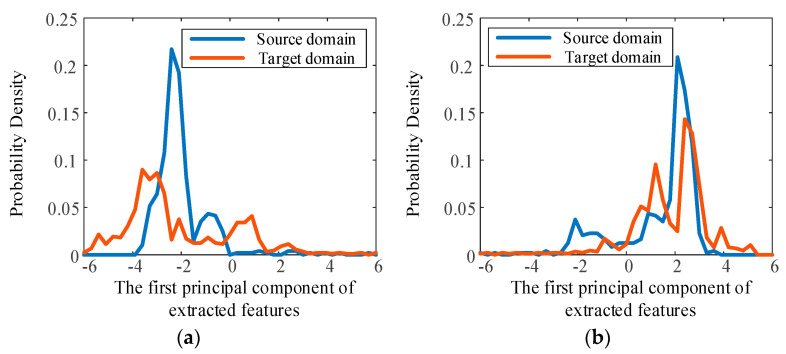
Probability density distribution of the first principal component of extracted features. Source domain is bearing3-1, target domain is bearing2-1. (**a**) Without transfer learning. (**b**) With transfer learning.

**Table 1 sensors-23-00227-t001:** Datasets used in experiments.

Operating Conditions	Radial Force (kN)	Rotating Speed (rpm)	Training Datasets	Test Datasets
**Condition2**	4.2	1650	Bearing2_1	Bearing2_2
**Condition3**	5	1500	Bearing3_1	Bearing3_3

**Table 2 sensors-23-00227-t002:** Parameters set of the network.

Layer	Size	Activation Function
Input	32 × 32	\
Convolutional layer 1	Kernel: 5 × 5 × 1 Stride: 3	ReLU
Pooling layer 1	Pool: 1 × 1 Stride: 1	\
Convolutional layer 2	Kernel: 5 × 5 × 6 Stride: 2	ReLU
Pooling layer 2	Pool: 1 × 1 Stride: 1	\
Net2	Layer 1: 160 × 10 Layer 2: 10 × 1	Sigmoid
Net3	Layer 1: 160 × 10 Layer 2: 10 × 1	Sigmoid

**Table 3 sensors-23-00227-t003:** Prediction metric results comparison.

Metric	Method	Bearing3_3	Bearing2_2
RMSE	Without trans.	27.23%	39.90%
	Proposed	7.01%	14.67%
MAPE	Without trans.	48.00%	82.2%
	Proposed	18.36%	28.10%
Precision	Without trans.	26.23%	20.19%
	Proposed	6.48%	12.63%

**Table 4 sensors-23-00227-t004:** RMSE prediction results under different noise levels.

Standard Deviation of Noise (g)	Bearing3_3
0.1	12.07%
0.2	17.22%
0.5	17.12%
1	17.00%
1.5	17.69%
2	26.51%

**Table 5 sensors-23-00227-t005:** Performance comparisons of related methods.

Method	Training Bearings ID		Test Bearings ID	Mean RMSE	Mean MAPE
	Source Domain	Target Domain			
DTMLKR [27]	Ber1_1,1_2	Ber2_1,2_2	Ber2_6	15%	34%
SCAE + MK-MMD [28]	Ber1_1,1_2	Ber2_1,2_2	Ber2_7	10.78%	14%
TLMLP [29]	Ber1_1,1_2,1_3, 1_4,1_5,1_6,1_7	Ber2_1,2_2	Ber2_6	29.83%	\
Proposed method	Ber3_1	Ber2_1	Ber2_2	14.67%	28.10%

## Data Availability

The raw vibration data used in the paper can be download from https://github.com/wkzs111/phm-ieee-2012-data-challenge-dataset.

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
