# Peer review of "Method for Predicting RUL of Rolling Bearings under Different Operating Conditions Based on Transfer Learning and Few Labeled Data"

_sensors, 2022, doi:10.3390/s23010227_

Round 1

Reviewer 1 Report

The paper proposed a transfer learning model, which solved the discrepancy of labels and features of the bearing operated under varying conditions to predict its remaining run-life. Overall, the paper’s content and topic are within the journal. However, it is subjected to a major revision before further consideration of possible publication on the journal.

1.     The first deficiency of the paper is poor written English, which surely brings down its academic value. For instance, lines 10, 31, 203 etc. are fragmental sentences, which are almost unreadable.

2.     Line 56, “Transfer learning, as a deep learning algorithm, provides a solution for different distribution prediction, which is widely used in various fields[9]”. The next literature review didn’t provide the detailed practical applications of the transfer learning algorithm.

3.     Section 3.1, the wavelet transformation was employed, but didn’t give brief introduction. Line 188, the sentence lacks causal relationship.

4.     Line 198, “the kernel width σ is set to 1”, what is the kernel? Please give more in-depth and detailed description of the adopted model.

5.     Line 224~226, is it appropriate to define the model precision by the dispersity of the errors?

6.     Fig 2, the bearing subfigure is improper here. Should provide the data plotting under different working conditions.

7.     Fig. 5 and 6, the real RUL versus time is a straight line. What on earth is the meaning of predicting RUL? Is it better to just apply least-square algorithm for curve fitting?

8.     To validate the robustness of the model, the Gaussian noise was imposed to the laboratory data. Is it possible to use directly extra field data to validate the model robustness?

Reviewer 2 Report

In the current study Rolling Bearing RUL Prediction Method Based on Transfer Learning and Few Labeled Data Used for Different Operating Conditions is investigated and studied. The overall structure of the paper is good. Manuscript is having lot off short form and symbols so, Authors need to check the abbreviations or nomenclatures throughout the manuscript. For example in introduction section EM, LSTM etc. not clearly mention. Better ways add the table for abbreviations or nomenclatures at stating of manuscript before introduction section.  

Reviewer 3 Report

See attachment.

Round 2

Reviewer 1 Report

The paper has addressed all my review comments in a good manner. I recommend it to be accepted by the journal.

Reviewer 3 Report

The authors have addressed satisfactorily the reviewer's comments. Hence, the paper is recommended for acceptance.